# Extracorporeal Gas Exchange for Acute Respiratory Distress Syndrome: Open Questions, Controversies and Future Directions

**DOI:** 10.3390/membranes11030172

**Published:** 2021-02-28

**Authors:** Marco Giani, Simone Redaelli, Antonio Siragusa, Benedetta Fumagalli, Roberto Rona, Giuseppe Foti

**Affiliations:** 1School of Medicine and Surgery, University of Milano-Bicocca, 20126 Milan, Italy; s.redaelli32@campus.unimib.it (S.R.); a.siragusa3@campus.unimib.it (A.S.); b.fumagalli@campus.unimib.it (B.F.); giuseppe.foti@unimib.it (G.F.); 2Emergency and Intensive Care, Azienda Socio Sanitaria Territoriale di Monza, 20900 Monza, Italy; r.rona@asst-monza.it

**Keywords:** acute respiratory distress syndrome, ARDS, veno-venous extracorporeal membrane oxygenation, ECMO, indications, prone positioning, mechanical ventilation, anticoagulation

## Abstract

Veno-venous extracorporeal membrane oxygenation (V-V ECMO) in acute respiratory distress syndrome (ARDS) improves gas exchange and allows lung rest, thus minimizing ventilation-induced lung injury. In the last forty years, a major technological and clinical improvement allowed to dramatically improve the outcome of patients treated with V-V ECMO. However, many aspects of the care of patients on V-V ECMO remain debated. In this review, we will focus on main issues and controversies on caring of ARDS patients on V-V ECMO support. Particularly, the indications to V-V ECMO and the feasibility of a less invasive extracorporeal carbon dioxide removal will be discussed. Moreover, the controversies on management of mechanical ventilation, prone position and sedation will be explored. In conclusion, we will discuss evidences on transfusions and management of anticoagulation, also focusing on patients who undergo simultaneous treatment with ECMO and renal replacement therapy. This review aims to discuss all these clinical aspects with an eye on future directions and perspectives.

## 1. Introduction

Acute respiratory distress syndrome (ARDS) is characterized by an acute and diffuse inflammatory lung injury of different etiologies which is associated to hypoxemic and, sometimes, hypercapnic respiratory failure [1]. Historically, veno-venous extracorporeal membrane oxygenation (V-V ECMO) improved gas exchange in acute respiratory failure (ARF) without increasing the probability of long-term survival, as first described by Zapol et al. in 1979 [2]. After this trial, ECMO in adult patients with ARF was nearly abandoned for many years. Meanwhile, ECMO was employed to treat hypoxemic newborns with greater survival benefit [3]. However, in the last decades, technological breakthroughs (e.g., improved oxygenators, heparin-coated surfaces) and enhanced management (e.g., trained personnel, fewer complications) allowed a significant improvement in survival of ARDS patients treated with V-V ECMO, leading to an increased utilization of this extracorporeal technique [4,5,6,7], particularly since the influenza pandemic in 2009 [8,9]. Yet, to date, V-V ECMO has never proved unequivocally to impact long-term survival. Modern protective ventilation strategies contributed to ameliorate outcomes [10,11,12] by preventing the ventilator-induced lung injury (VILI) [13]. Thus, they became the cornerstone of ARDS treatment [14,15,16]. VILI is an acute lung injury inflicted by mechanical ventilation and recognizes multiple causes, as alveolar overdistension, excessive positive pressure ventilation and continuous opening and closing of alveoli and inflammation. Regrettably, VILI is not always avoidable, particularly in case of very low respiratory system compliance [17]. In such circumstances, V-V ECMO remains an appealing approach, because it makes protective ventilation feasible. In some circumstances, particularly when ARDS is associated to severe pulmonary hypertension or cardiac dysfunction for sepsis, veno-arterial ECMO (V-A ECMO) is required for both gas exchange and cardiac support. However, V-A ECMO support is beyond the scope of this review and will not be discussed.

During the current Covid19 pandemic, V-V ECMO has been extensively used as an advanced therapy in patients with ARDS, refractory to conventional treatment [18,19]. However, due to a steep increase in ARDS incidence, ECMO requirements may overcome the availability of ECMO-capable facilities. Thus, within this healthcare emergency, shortage of personnel and hospital capabilities determines significant challenges and require appropriate patients selection, not to overburden an already stressed healthcare system [20].

Despite an irrefutable improvement of outcome in the last decades [2,21], a lot has still to be done to reduce mortality and to improve quality of life after ECMO. The scope of this review is to discuss the open questions and controversies in the management of ARDS patients supported with V-V ECMO, with an eye on future directions and perspectives. Relevant interventional and observational studies are analyzed to focus on the main issues and controversies currently affecting outcomes in ARDS patients on V-V ECMO. Table 1 summarizes published and ongoing randomized controlled trials on V-V ECMO in ARDS patients relevant to the topics treated in this review.

**Table 1 membranes-11-00172-t001:** Published and ongoing randomized controlled trials on VV-ECMO in ARDS patients. Trials are grouped by the research question they address.

Trial Name [Status]	Main Enrollment Criteria	Enrolled Patients (N)	Interventional Group	Control Group	Primary Endpoint	Results
**Survival**						
CESAR (2009) [has results] [22]	Severe, but potentially reversible respiratory failure defined as: Murray score >2.5 or hypercapnia with pH < 7.20	180	ECMO consideration and potential initiation after transport to an ECMO capable facility	Conventional treatment	Death or severe disability * at 6 months	RR (95% CI) 0.69 (0.05 to 0.97); (*p* = 0.03)
EOLIA (2018) [has results] [21]	Severe respiratory failure defined as: P/F < 50 mmHg for > 3 h or P/F < 80 mmHg for > 6 h or pH < 7.25	249	ECMO	Conventional treatment	Death at 60 days	RR (95% CI) 0.76; (0.55 to 1.04); (*p* = 0.09)
**ECCO2R**						
Extracorporeal Carbon Dioxide Removal for Acute Respiratory Distress Syndrome (NCT00000572) [completed]	ARDS w/PaO2 < 50 mm Hg for three times	40 (estimated)	Detailed Electronic Protocol Controlled ECCO2R w/reduced positive-pressure ventilation	Detailed Electronic Protocol Controlled positive-pressure ventilation	Death at 30 days	-
REST (NCT02654327) [active, not recruiting]	Respiratory failure with P/F < 150	1120 (estimated)	ECCO2R to enable lower tidal volume mechanical ventilation	Conventional treatment	Death at 90 days	-
**Timing of ECMO initiation**						
ECMO-VID (NCT04341285) [not yet recruiting]	Covid19 respiratory failure w/P/F < 100 mmHg	200 (estimated)	ECMO w/in 24 h of ICU referral	ECMO as rescue after failure of conventional treatment	Death at 28 days	-
ELIEO-Trial (NCT04208126) [not yet recruiting]	ARDS with P/F < 200 mmHg	200 (estimated)	ECMO at ICU admission	Conventional treatment. ECMO as rescue treatment allowed	Death at 28 days	-
**Mechanical ventilation**						
New Lung Ventilation Strategies Guided by Transpulmonary Pressure in VV-ECMO for Severe ARDS [has results] [23]	Patients with V-V ECMO for ARDS	104	ECMO + transpulmonary pressure ventilation	ECMO + conventional ventilation	Proportion of weaned patients from V-V ECMO	71.2% vs. 48.0%; (*p* = 0.017)
Low Frequency, Ultra-low Tidal Volume Ventilation in Patients with ARDS and ECMO (NCT03764319) [recruiting]	Moderate to severe ARDS + ECMO	40 (estimated)	ECMO + ultraprotective ventilation	ECMO + conventional ventilation	Ventilator free days	-
**VILI**						
ECMO-VILI (NCT03918603) [recruiting]	ARDS with P/F < 70 mmHg for > 2 h or 70 < P/F < 100 mmHg w/Ppl > 35 cmH_2_O and pH < 7.20	30 (estimated)	ECMO + prone positioning	ECMO + usual care	Inflammation of biotrauma (interleukine dosage)	-
**Prone positioning**						
Early Use of Prone Position in ECMO for Severe ARDS (NCT04139733) [recruiting]	ARDS with P/F < 80 mmHg or pH < 7.20 and paCO_2_ > 60 mmHg	110 (estimated)	ECMO + prone position	ECMO + supine position	VV-ECMO duration time	-
PRONECMO (NCT04607551) [not yet recruiting]	Severe ARDS	170 (estimated)	ECMO + prone position	ECMO + supine position	Time to successful ECMO weaning w/in 60 days following randomization	
**Anticoagulation**						
A-FREE ECMO (NCT04273607) [not yet recruiting]	Adult patient with ARDS on V-V ECMO	40 (estimated)	ECMO without anticoagulation	ECMO + anticoagulation w/UFH	ECMO associated thrombotic complications	-
TEG Anticoagulation Monitoring During ECMO [has results] [24]	Patients with acute respiratory failure with ECMO	42	Anticoagulation management based on TEG	Anticoagulation management based on aPTT	Safety (n° of hemorrhage, thrombosis, transfusions)	No differences between groups
BIV-ECMO2 (NCT03965208) [recruiting]	Adult patients on ECMO	34 (estimated)	Anticoagulation w/bivalirudin	Anticoagulation w/UFH	Percentage of time in the target anticoagulation range	-
GATRA study [has results] [25]	Patients on V-V ECMO for respiratory failure	48	ATIII supplementation	No ATIII supplementation	UFH dose to maintain aPTT ratio between 1.5–2	No difference in UFH dose between groups

* severe disability is defined as ‘confined to bed’ or ‘unable to wash and dress’. ARDS is defined according to the Berlin ARDS definition (2012), if not otherwise specified [1]. Abbreviations: V-V ECMO, veno-venous extracorporeal membrane oxygenation; RR, relative risk; CI, confidence interval; ECCO2R, extracorporeal carbon dioxide removal; ARDS, acute respiratory distress syndrome; P/F, ratio of arterial oxygen partial pressure to fractional inspired oxygen; Ppl, plateau pressure; VILI, ventilation induced lung injury; w/, with; w/in, within; paCO_2_, partial arterial pressure of carbon dioxide; UFH, unfractionated heparin; TEG, thromboelastography; aPTT, activated partial thromboplastin time; ATIII, antithrombin III.

## 2. Evidence on V-V ECMO Use in ARDS

After many years of inconclusive trials and scientific discussion, two important studies have changed the evidence on V-V ECMO use.

About 10 years ago, the CESAR trial [22] clearly showed that the most severe ARDS patients should be transferred to an ECMO-capable center to significantly improve survival without severe disability. Even if only 75% of patients actually received ECMO, it is highly likely that the use of ECMO had an impact on the survival benefit.

The more recent EOLIA trial [21] randomly assigned 249 patients with severe ARDS to receive early V-V ECMO or conventional tidal volume (Vt) and pressure limited ventilation (including late ECMO as rescue therapy) (Table 1). The primary endpoint was mortality at 60 days. However, according to pre-specified futility rules, the trial was stopped early at 75% of recruitment, because of lack of difference in mortality at 60 days between groups. This probably made the trial underpowered to address the research question. In spite of inconclusive survival results (35% and 46% mortality in ECMO and control group, respectively, *p* = 0.09), the high percentage of sicker patients that crossed over from the conventional treatment group to the ECMO group for rescue therapy (28%) endorsed the use of V-V ECMO in life-threatening hypoxemia. Moreover, post-hoc Bayesian analysis and individual patient data meta-analysis provided more favorable interpretation of the study results. In details, post-hoc Bayesian analysis on the EOLIA trial, suggested a probability of ECMO success in decreasing mortality of 88% to 99% according to the chosen priors. Similarly, the analysis of data of the 429 patients enrolled in the EOLIA and CESAR trial showed a relative risk of death at 90 days of 0.75 (95% confidence interval 0.6–0.94, *p* = 0.013, I^2^ = 0%) in the ECMO group compared to conventional management [21,26,27].

## 3. Indications and Counterindications for V-V ECMO in ARDS Patients

ECMO represents one of the most invasive procedures for respiratory failure in critically ill patients. As per current ELSO guidelines [28], actual indications for V-V ECMO in ARDS include ensuring vital oxygenation in patients with extreme hypoxia due to ARF, a condition associated with a 50–80% risk of mortality. V-V ECMO is an indisputable life-saving therapy when extreme hypoxia persists after failure of other less invasive rescue therapies (i.e., prone positioning and inhaled nitric oxide). Another indication to ECMO in ARDS is carbon dioxide retention despite mechanical ventilation with high plateau pressure (>30 cmH_2_O).

V-V ECMO provides lung rest by reducing the burden of mechanical ventilation on the sick lung (i.e., reduction of ventilator-induced lung injury, VILI) and a strong physiological rationale supports its utilization when very high pressures and/or tidal volumes are required to maintain vital gas exchange. However, it is unclear how to define the oxygenation and ventilation load cutoffs which mandates the start of the V-V extracorporeal support. To subside this issue, the Murray score, which consider oxygenation, ventilation and respiratory mechanics parameters has been proposed [29]. This tool may be used to select candidates to ECMO and was an enrollment criteria of CESAR trial [22] (Table 1).

On the contrary, there is no specific contraindication to ECMO. However, long-lasting not protective mechanical ventilation, immunosuppressive status, recent or active central nervous system hemorrhage, terminal malignancy, nonrecoverable comorbidity are associated with poor outcome.

Also, increasing age is known to be associated with worse outcome and is thus included in the most common ECMO prognostic scores (the RESP score [30] and PRESERVE score [31]). An upper age limit has not been defined, as outcome in the elderly patients (i.e., 60 to 70 years old) varies largely among different centers [32,33,34]. Rather than a fixed age limit (i.e., 65 years old), it seems to be advisable to include age, comorbidities and performance status into a comprehensive patient evaluation before providing such an invasive treatment.

## 4. V-V ECMO Versus Extracorporeal Carbon Dioxide Removal (ECCO2R) in ARDS

Veno-venous extracorporeal support may be used either as a rescue therapy for hypoxia [2], or to decrease ventilatory load and potential ventilation induced lung injury (VILI) by the extracorporeal removal of carbon dioxide (CO_2_) (ECCO2R) [35]. With the first goal, a high-flow V-V ECMO (3–5 l/min of extracorporeal blood flow) is required to achieve adequate blood and organs oxygenation. Contrarily, CO_2_ clearance requires only a low-flow extracorporeal support (i.e., 500 to 1500 mL/min of extracorporeal blood flow). Eventually, this prevents respiratory acidosis and allows the reduction of ventilatory burden. This is determined by the different physiology of oxygen and CO_2_ exchange through the membrane lung (ML). Oxygen transfer mainly depends on the extracorporeal blood flow, whereas CO_2_ transfer depends on the sweep gas flow rate at the ML [36]. Commonly, in ARDS patients requiring extracorporeal support, high flow V-V ECMO is the technique of choice and hypoxemia is the main inclusion criteria in extracorporeal support clinical trials [2,21,22,37]. However, Gattinoni et al. questioned the idea that a very low arterial oxygen tension determines tissue hypoxia. Indeed, patients with an arterial partial pressure of oxygen lower than 60 mmHg, may have no organ damage, but the lung [36]. Therefore, usually, high flow V-V ECMO is started based on the clinical judgement.

Besides, if the extracorporeal support is prompted to decrease the ventilatory load, a less invasive ECCO2R technique may be a reasonable choice. Nevertheless, a recent trial on ECCO2R and ultra-protective lung ventilation, used an oxygenation index and not ventilatory load as an inclusion criterion [38].

The combined use of ECCO2R and mechanical ventilation has proved to be feasible when compared to mechanical ventilation alone, however data on clinical outcomes are lacking [38,39]. The ongoing REST trial (NCT02654327) aims to establish, whether ECCO2R and lower Vt improve all-cause mortality and it is cost-effective in comparison with standard of care.

ECCO2R instrumentation limits the maximum blood flow achievable, becoming useless if the patient worsens and develops life-threatening hypoxemia. Indeed, in a study of ECCO2R safety, prone positioning and high flow V-V ECMO were required as rescue therapy for life-threating hypoxemia in two and four out of 15 patients, respectively [39]. Further research is warranted to determine the safety and feasibility of a pure ECCO2R technique versus a high flow V-V ECMO.

Compared to “full” V-V ECMO, CO_2_ removal systems have been aimed at low invasiveness, through low extracorporeal blood flows and smaller cannulae. However, with the current technology, a blood flow of 750 to 1000 mL/min is required to achieve an efficient CO_2_ removal [40]. To further decrease the extracorporeal blood flow, Zanella et al. previously developed an experimental ECCO2R technique based on acidification of blood entering the membrane lung (Acid Load CO_2_ removal, ALCO2R), which allows to increase CO_2_ removal by converting bicarbonate ions into dissolved CO_2_ [41,42,43]. More recently, the respiratory electrodialysis has been described. Through an hemofilter and an electrodialysis cell, blood electrolytes are modulated to convert bi-carbonates to CO_2_ before entering the ML, enhancing ML CO_2_ extraction [44,45]. Further research is warranted to verify the clinical applicability of these innovative techniques.

Moreover, in the next future, clinical trials should clarify the indications of low-flow ECCO2R techniques and determine whether their use may lead to an improvement in clinical outcomes.

## 5. Mechanical Ventilation in ARDS Patients on V-V ECMO

While the objective of mechanical ventilation in ARDS patients on V-V ECMO is to reduce VILI and to promote lung healing, no consensus exists on the best ventilation strategy. A recent multicenter prospective cohort study, showed that V-V ECMO allows for Vt reduction (6.4 ± 2.0 to 3.7 ± 2.0 mL/kg predicted body weight, *p* < 0.001) and Plateau Pressure (Ppl) decrease (32 ± 7 to 24 ± 7 cmH_2_O, *p*< 0.001) compared with pre-ECMO settings. Moreover, driving pressure (DP) fell from 20 ± 7 to 14 ± 4 (*p* < 0.001) [46]. All of these parameters are well-known predictors of mortality [10,12,47], even in patients supported with V-V ECMO [48]. Notably, in the LIFEGARDS study, ventilator settings during V-V ECMO had no impact on survival, but patients homogenously received protective ventilation compared to pre-ECMO settings [46].

However, Del Sorbo et al. noticed that even a low tidal volume, low plateau pressure mechanical ventilation may cause VILI. Indeed, decreasing the DP to zero is associated to reduced plasma concentrations of inflammatory biomarkers, but patient-centered outcomes have to be tested [49].

As V-V ECMO allows for reduction in Vt, atelectasis could increase and determine atelectrauma. Hence, positive end expiratory pressure (PEEP) is used to maintain alveolar recruitment, but simultaneously, alveolar overdistension should be avoided. The mean reported PEEP in V-V ECMO patients is 11 ± 3 cmH_2_O [46], consistent with a recruited lung strategy. As for the setting of PEEP, ELSO guidelines recommend a PEEP as high as tolerated [28], while the Consensus Conference on ECMO in ARDS favors the minimum PEEP [50]. Different approaches permit a tailored PEEP setting: choosing the PEEP associated with the best compliance of the respiratory system [12] or keeping transpulmonary pressure (Ptp) between 0 and 10 cmH_2_O [51]. Indeed, a Ptp-guided ventilation, increased the probability of successful weaning from ECMO [23] (Table 1) compared to ELSO guidelines strategy [28]. Moreover, pressure-volume curve [52] and electrical impedance tomography [53] may be used to set PEEP.

To combine all the ventilation parameters which may cause VILI, the mechanical power has been proposed [54]. This parameter includes variables which have been associated with lung damage: tidal volume, flow, driving pressure, respiratory rate, and PEEP. A mechanical power higher than 17 Joule per minute represents an increased risk of death [55]. Notably, V-V ECMO allowed for a 75% and 66% mechanical power reduction in the LIFEGARDS [46] and EOLIA [21] population, respectively, with the mechanical power being above the safety threshold during the pre-ECMO setting.

The majority of patients on V-V ECMO receives controlled mechanical ventilation and pressure-targeted modes are chosen the most [46]. However, in the recovery phase of ARDS, assisted-modes may allow for respiratory muscles training and weaning [56]. On the contrary, a poorly applied spontaneous ventilation could determine patient self-induced lung injury, through asynchronies and regional overdistension [57]. An experimental animal study showed spontaneous breathing with low respiratory efforts and Vt did not increase lung injury compared with near-apneic ventilation [58]. Moreover, neurally adjusted ventilatory assist ventilation has been associated with fewer ventilator asynchronies in V-V ECMO patients [59].

In a nutshell, V-V ECMO permits a protective ventilation strategy and commonly a rest lung strategy is adopted [46,60]. However, which level of rest and the role of assisted breathing on V-V ECMO need clarification. A trial to assess if a continuous positive airway pressure strategy mitigates VILI in comparison to tidal ventilation is ongoing (NCT01990456).

## 6. Prone Positioning during V-V ECMO

Prone positioning (PP) has become a standard of care in mechanically ventilated patients with severe ARDS. PP allows a significant improvement in oxygenation and prevents VILI by contrasting its mechanical (i.e., inhomogeneity of tidal volume distribution) and nonmechanical (e.g., healthy lung contamination) determinants [61]. CO_2_ clearance may be improved, and CO_2_ reduction after PP is associated with better outcome [62]. Most important, after more than a decade of scientific debate, a multicenter randomized trial by Guérin et al. [63] showed that early and prolonged prone positioning determines a survival benefit in the most severe patients (i.e., with a ratio of arterial oxygen partial pressure to fractional inspired oxygen below 150 mmHg).

Research data show that the more severe is ARDS, the greater is the benefit of PP on outcome [64]. For this reason, continuing PP in the most severe ARDS patients after start of V-V ECMO support seems to have a strong rationale. The use of PP during V-V ECMO support is feasible [65,66], does not cause any cannula disfunction or displacement [67], and it is associated with few adverse events [68,69]. A physiologic study by Franchineau and coll. [70] confirmed the potential of this procedure to improve lung mechanics and reduce VILI. Furthermore, some preliminary evidence [69,71] suggests that the use of PP during ECMO is associated with improved survival. However, the results of retrospective studies may be biased, as this procedure in some centers may be limited to the most severe patients. This may explain the conflicting findings of other studies [72,73].

## 7. Sedation during V-V ECMO

Sedation management in critically ill patients has been revised in the last years. Recent guidelines strongly recommend light sedation and daily interruption of sedation because this approach showed to be associated with shorter duration of mechanical ventilation and shorter length of stay in ICU [74,75].

However, no specific guidelines exist for ECMO patients and the approach used is variable among different ECMO centers [76]. Some specific features of patients on V-V ECMO (e.g., the very low compliance of the respiratory system) make a light sedation approach difficult.

In fact, prolonged periods of deep sedation are often required for the most severe cases [77]. Achieving and maintaining a deep sedation plan may be a difficult goal for the clinician [78]. Midazolam and propofol are the most used sedatives, but they bring some well-known adverse effects such as accumulation [79], muscular toxicity [80] and tachyphylaxis [81]. Moreover, pharmacokinetics of intravenous sedatives may be affected by the extracorporeal circuit [82]. Volatile anesthetics may represent a valuable alternative to intravenous drugs [83]. Despite their widespread use for anesthesia, they are seldom used in the critical care setting [84,85].

Volatile sedation may be feasible even during ultraprotective ventilation and V-V ECMO. The low minute ventilation of V-V ECMO patients allows to reach the volatile gas concentration target (and the sedation target) with low infusion rates [83,86], potentially reducing its economic impact. Additional investigation may clarify if volatile sedation may provide any clinical benefits in ECMO patients requiring prolonged sedation.

Whenever the clinical condition allows it, current evidence [74,75] strongly support the achievement of a minimal sedation plan. A retrospective study [78] included 45 ARDS patients treated with V-V ECMO and evaluated their sedation management during ECMO support. The authors found that, although in the first phase after cannulation a deep sedation approach was used, 78% of patients achieved at least one day of light or intermediate sedation, during their ECMO course.

A light sedation protocol also allows an early mobilization, which seems feasible and safe in ECMO patients. Indeed, a prospective observational study showed up to 37% of patients with ECMO received some degree of active mobilization. The majority of them was transferred from bed to chair or marched on the spot [87]. In a retrospective study on 100 patients on ECMO support, 35% of patients received active physical therapy with a median delay from ECMO initiation of two days. Moreover, 18% of patients ambulated a median distance of 53 m [88]. Further studies are needed to evaluate if early physical therapy could improve outcomes in ECMO patients, especially during the ECMO-weaning process.

## 8. Hemoglobin Threshold for Transfusion during V-V ECMO

Hemoglobin (Hb) threshold and strategies on red blood cells (RBC) transfusion are some of the most debated aspects of patients’ management during V-V ECMO.

Hb is the main factor affecting the arterial oxygen content and, consequently, the oxygen delivery (DO_2_). Increasing Hb through RBC transfusion is a widespread strategy used to improve DO_2_ during ECMO support [89]. Furthermore, patients on ECMO often show increased transfusion requirements due to a high rate of bleeding events caused by anticoagulation.

RBC transfusion is not free from adverse effects. Although the risk of infections has been extremely reduced in the last years, noninfectious transfusion-associated adverse events have become increasingly known. These are attributable to the contaminants of the blood products, and to the alterations of RBC during the storage. They are usually classified as immune mediated reactions (i.e., hemolytic transfusion reactions, transfusion-related acute lung injury, transfusion-related immunomodulation, etc.) and nonimmune mediated reactions (i.e., nonimmune hemolysis, transfusion-associated circulatory overload, metabolic and coagulopathic complications, etc.) [90].

In a previous study, Martucci et al. [89] showed that ECMO patients who received more RBC units (>150 mL/d) had a lower survival compared to patients who received less RBC units (62.7% vs. 89.9%, log-rank *p* < 0.01). However, no causative effect can be drawn by this association.

Current guidelines lack specific recommendations for RBC transfusion during ECMO due to low quality evidence. Actual evidence from critical non-ECMO patients suggests that a lower Hb transfusion threshold (i.e., 7 g/dL) is safe and reduces the risk of transfusion-associated adverse events [91].

Retrospective observational studies suggest that such restrictive strategy could be applied to ECMO patients without affecting clinical outcomes. Voelker et al. reported the feasibility of maintaining Hb levels between 7 and 9 g/dL in V-V ECMO, achieving survival rates comparable to the ELSO registry cohort [92]. This strategy may significantly reduce the need of transfusion without affecting outcome [93].

The TRAIN-ECMO survey [89] assessed that only 46% of centers used the Hb level alone as a transfusion trigger. Notably, in addition to Hb level, other parameters were used to evaluate the need for RBC transfusion, such as oxygenation indexes (e.g., mixed venous and arterial saturation and oxygen delivery), ongoing bleeding and tissue perfusion variables (e.g., lactates and clinical signs of hypoperfusion).

A restrictive transfusion strategy in V-V ECMO patients seems feasible. However, to date, no “magic number” can be considered as a universal transfusion threshold in this population. Therefore, clinicians should always use an integrated physiologic approach to determine the patient need for RBC transfusion, weighting the benefits and the known risks.

## 9. Anticoagulation

Anticoagulation during ECMO support is essential to reduce the risk of thrombotic complications, which are reported in about 15% of V-V ECMO patients [94,95] On the other hand, bleeding events are among the main contributors of mortality in these patients [96]. For this reason, optimizing the management of anticoagulation seems crucial to improve outcome of ECMO patients. Alterations of hemostasis during ECMO occur as a consequence of multiple mechanisms: direct contact between blood and non-biological surfaces, shear stress and patient factors. Interestingly, the same mechanisms also determine bleeding tendency [97]. In the last decades, these alterations of hemostasis were attenuated by the widespread diffusion of heparin-coated circuits and new centrifugal pumps, which allow to reduce the activation of coagulation, hemolysis and anticoagulation requirements [98,99]. To date, no clear recommendations exist about anticoagulation management because of low quality evidence. In fact, ELSO anticoagulation guidelines simply report the available strategies and approaches [28,100].

### 9.1. Anticoagulant Drugs

Unfractionated heparin (UFH) is the systemic anticoagulant of choice in most ECMO centers [101]: it is inexpensive, it has a short half-life and it is reversible. Nevertheless, direct thrombin inhibitors (DTI) have some potential advantages: they act directly on both clot-bound and thrombin independently from antithrombin, have more predictable pharmacokinetic properties and can be used in patients with heparin-induced thrombocytopenia (HIT). Among direct thrombin inhibitors, bivalirudin and argatroban has been most used in extracorporeal applications. To date, there is no clear indication for direct thrombin inhibitors use during V-V ECMO. Further studies will have to assess the role and potential benefit of these drugs in patients without heparin-induced thrombocytopenia.

### 9.2. Anticoagulation Monitoring

To date, there is no standardization about monitoring of the unfractionated heparin effect and ELSO guidelines simply suggest a multimodal approach [28]. The most used methods are activated clotting time (ACT) and activated partial thromboplastin time (aPTT), both of which have limitations.

Activated clotting time has been considered for decades as the gold standard method in extracorporeal treatments. It is a whole blood test that evaluates the global hemostasis function with no standardized target. However, in ECMO patients, ACT is usually maintained between 160 and 220 s [28,102]. It is a point-of-care test (POCT), having shorter turnaround times for results. On the other hand, being a global evaluation of hemostasis, it can be influenced by multiple factors other than unfractionated heparin(e.g., platelet count and function, hypothermia and coagulation factors deficiency). In addition, ACT has poor correlation with heparin concentration at the lower dosage used during V-V ECMO, compared with cardiac surgery setting [103,104].

The activated partial thromboplastin time is a plasma-based assay first described in the 1950s as a test for hemophilia [105]. Later, it has become the standard to monitor unfractionated heparin effect. In V-V ECMO patients, the normally used aPTT ratio range is 1.5–2.0 (i.e., 1.5 to 2 times the normal aPTT). Being a plasma-based assay, it is not influenced by platelet count and function, but it is by other factors such as deficiency of coagulation factors, fibrinogen or von Willebrand factor [106]. Compared to activated clotting time, aPTT seems more accurate at the low unfractionated heparin doses used in V-V ECMO patients. A systematic review [107] determined an aPTT based strategy was associated with fewer bleeding and thrombotic events during V-V ECMO compared to ACT. Moreover, the authors found that a lower target of aPTT (i.e., <60 s) was associated with fewer life-threating bleeding events.

Limitations of these two tests led some centers to prefer the anti-Xa, a more specific assay that estimates unfractionated heparin concentration by measuring the level of inhibition of factor Xa. Anti-Xa assay, being a pharmacokinetic test, has a better correlation with heparin dose compared to pharmacodynamic tests (ACT, aPTT) [108]. ELSO guidelines suggest a target of 0.3–0.7 IU/mL. A systematic review and meta-analysis [109] showed that an anti-Xa based approach was significantly associated with fewer bleeding episodes and lower mortality compared to a management based on time-guided assays, without increasing the risk of thrombotic events. The main limitation of the anti-Xa assay is that it does not evaluate the global hemostatic state [110]. Patients with similar anti-Xa level may present very different concentration of coagulation factors (e.g., a hypercoagulable patient versus a hemorrhagic patient with diluted clotting factors), resulting in very different risks of thrombosis or bleeding.

Recently, there has been growing interest in viscoelastic point-of-care test, such as thromboelastography (TEG) and rotational thromboelastometry (ROTEM). These tests provide information on the whole hemostatic process, describing clot formation, strength and lysis. In addition to their utility in the understanding of hemostasis during ECMO [100], TEG reaction time “R” has been proposed to be used to monitor UFH anticoagulation. TEG analysis suggest that many ECMO patients monitored by aPTT may receive excessive anticoagulation [111]. A recent pilot randomized controlled trial included 42 patients treated with V-V ECMO for acute respiratory failure and compared a TEG-based anticoagulation management and a standard aPTT-based approach. TEG-based protocol was safe and feasible, allowing a reduction of heparin dose (11.7 vs. 15.7 IU/kg/h, *p* = 0.03) without increasing thrombotic events. Moreover, a nonsignificant reduction of bleeding episodes (48% vs. 71%, *p*= 0.21) was recorded (Table 1). ROTEM was less studied in this setting [24]. In a recent prospective observational study by our research group, we found a poor correlation between TEG reaction time and aPTT, whereas a moderate correlation was found between ROTEM CT and aPTT. Further prospective studies are needed to assess the role of viscoelastic point-of-care test to monitor anticoagulation in ECMO patients [112].

### 9.3. Antithrombin

Heparin exerts its anticoagulant activity interacting with antithrombin (also termed antithrombin III). For this reason, if plasmatic antithrombin activity is reduced, supplementation is commonly used to increase UFH efficacy during ECMO support. A recent survey [101] found that antithrombin was routinely monitored in 49% and/or routinely supplemented in 38% of the participants centers, despite a low evidence supporting its supplementation during V-V ECMO support. In the GATRA study [25], a randomized controlled trial by Panigada and coll., routine antithrombin supplementation did not decrease heparin requirements nor the incidence of bleeding and/or thrombosis in adult patients supported with V-V ECMO. However, patients in the control group showed relatively high AT III levels (Table 1). Future research will have to verify if antithrombin supplementation is indicated in specific patient subgroups (e.g., patients with very low AT III levels or heparin resistance).

### 9.4. Level of Anticoagulation

The level of anticoagulation required during V-V ECMO is a subject of debate. The ideal anticoagulation therapy should minimize the risks of bleeding and thrombosis. A recent retrospective study by Stokes and coll. [94] evaluated bleeding and thrombotic events in 55 adult patients on V-V ECMO receiving UFH anticoagulation. Bleeding events were more frequent and were associated with worse outcomes, whereas thrombotic episodes were not. Some authors suggest that level of anticoagulation currently in use may be too high, considering the previously described technological improvements. Two retrospective analysis [113,114] evaluating feasibility of V-V ECMO without systemic anticoagulation have been published. Prophylactic anticoagulation alone was safe and associated with reduced bleeding complications and transfusion requirements, without increasing the risk of thromboembolic complications.

These findings question the utility of the level of anticoagulation routinely used in most ECMO centers and lay the foundations for a randomized trial identifying an adequate anticoagulation management in V-V ECMO patients.

### 9.5. Anticoagulation of the Renal Replacement Therapy Circuit during V-V ECMO

Critically ill patients undergoing V-V ECMO frequently develop acute kidney injury, often requiring continuous renal replacement therapy (RRT) [115,116].

We already discussed that the contact of blood with nonbiological surfaces activates the coagulation cascade. Thus, in the presence of two extracorporeal circuits, the choice of an adequate anticoagulation strategy becomes essential.

Regional citrate anticoagulation has become the standard of care in CRRT circuits: citrate prevents clotting by chelation of ionized calcium, a fundamental cofactor of coagulation [117]. This technique, besides reducing the risk of bleeding connected with systemic anticoagulation, improves filters lifespan and lowers costs compared with systemic anticoagulation.

Despite its recognized advantages, the use of this technique rises concerns about the risk of electrolyte and acid-base disorders. Anyway, these complications are quite uncommon when strict adherence to the regional citrate anticoagulation protocol and accurate monitoring of the procedure are ensured by an adequately trained staff [118]. On the other hand, systemic anticoagulation is mandatory during ECMO support. Regional citrate anticoagulation cannot be used for ECMO, as the slow metabolism of citrate limits its use to blood flows below 150–200 milliliters per minute, narrowing its application to the field of CRRT.

When CRRT is combined with ECMO, regional citrate anticoagulation is rarely employed [119]. However, clotting of the CRRT filter with the sole systemic anticoagulation is frequent, due to low blood flow and lack of coating of CRRT surfaces. This may lead to activation of the coagulation cascade and consumption of platelets and clotting factors. A recent study by our group on V-V ECMO patients with renal failure, retrospectively assessed the feasibility of additional regional citrate anticoagulation on the CRRT circuit in patients receiving systemic heparinization [120]. The combination of regional citrate anticoagulation with systemic heparin allowed a significant reduction in CRRT circuit clotting compared to systemic heparinization alone (despite a higher aPTT in the latter group). Moreover, the addition of regional citrate anticoagulation was associated with reduced platelet consumption and lower dimers level. Therefore, we hypothesize that this approach may minimize the impact of the CRRT treatment on the delicate coagulative balance of these patients, potentially reducing the risk of hemorrhagic and thrombotic complications. However, these findings must be confirmed by prospective research.

## 10. Outlook

Although the basics of ECMO have remained unchanged since its introduction, its widespread use has challenged researchers to ameliorate ECMO equipment and to investigate on new technologies. ECMO systems are now smaller in dimension and the development of polymethylpentene fibers [121] has allowed for the introduction of low resistance, long-lasting and more biocompatible exchange membranes, although still imperfect. Moreover, the ECCO2R technique has the potential to decrease the ventilatory burden on the sick lungs, with less invasiveness compared to “full” ECMO, especially when applied through a single double-lumen cannula. As previously described, the respiratory dialysis could increase ECCO2R performance [44], making this technique more efficient. Therefore, in the next years, research should concentrate on developing more and more efficient and biocompatible membranes, miniaturized and less invasive equipment, which could help clinicians to ease the management of such complex patients and potentially improve the quality of care.

## 11. Conclusions

Despite a great clinical and technological improvement in the last decades, a lot has still to be done to further improve the outcomes of ARDS patients treated with extracorporeal support. Controversies exist in many topics. Indeed, the most appropriate setting of mechanical ventilation is still unknown. Whether a high PEEP is better than a low PEEP or a ventilated natural lung is preferred to a completely rested natural lung is a matter of debate. The ECCO2R technique could be a less invasive alternative to “full” ECMO, but which patients would benefit more is undetermined. Moreover, as prone positioning is safe and feasible in ECMO patients [69], is it appropriate for all? Which is the adequate level of sedation and the best sedative drugs? Also, the optimal anticoagulation management which reduces thrombosis and does not increase bleeding, along with the appropriate Hb threshold for transfusions have to be clarified. Future research should aim to resolve these issues and assess how they could affect outcome through randomized controlled trials, which are, so far, lacking.

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
