# Peer review of "Extracorporeal Gas Exchange for Acute Respiratory Distress Syndrome: Open Questions, Controversies and Future Directions"

_membranes, 2021, doi:10.3390/membranes11030172_

Round 1

Reviewer 1 Report

I would like thank the editors of membranes for the opportunity to review this review article. Giani et al. provide a literature overview of extracorporeal membrane oxygenation in ARDS patients. In times of COVID and the increase of multimorbid patients this topic is of high relevance. A review should allow the reader to get a good overview on the topic. Unfortunately, Giani et al miss to provide all basic information that the average reader needs to understand the article. This review has serious flaws, language has to be improved, it lacks details on the topic and, therefore, needs major revision.

My comments

  • In general, the authors mention various relevant topics (ARDS, VILI, NISHOT, ECCO2R etc.) and expect the reader to be up-to-date in these specific fields. I recommend to provide all needed basic information and bring all readers up-to-date before going deep.
  • Which literature did you choose? Please define which studies were included into your literature research (e.g. Top-Journals/prospective randomized trials/retrospective …) and which were excluded.
  • Please add information (+ a figure) of the number of published literature within the last decades since the invention of ECMO
  • I recommend to use another review (about ECMO) from membranes or a comparable journal as “template”
  • A short explanation or paragraph of VILI and ARDS should be added to the introduction, since this is the title of the manuscript. In general, the most common indications should be listed as well.
  • Although V-V ECMO might be clear to most readers, the introduction would benefit from a short paragraph about the history of ECMO, the difference between V-V- and V-A ECMO and what the most common indications are. Furthermore, a short note about the relevance of V-V ECMO in the current COVID pandemic should be part of the review as well.
  • When was the first V-V ECMO described? Why was it developed or who used it if the trials were inconclusive? What were the milestones hat improved the results? Technology?
  • Do not use informal English as e.g. it’s (use it is)
  • “Moreover, per-protocol and post-hoc Bayesian analysis provided more favor-60 able interpretation of the study results” This information is too rudimental and needs more explanation.
  • “Indications for V-V ECMO in ARDS patients”: This paragraph reads very unfinished and the text does not flow (general remark). In addition, “Age” cannot be an indication for ECMO, rather a contraindication, therefore it does not correlate to the heading of this paragraph.
  • Line 156: Please define J/min
  • I would recommend to add a list of abbreviations
  • All abbreviations must be explained at first use. (e.g. line 183: P/F ratio)
  • Which implantation techniques are available for V-V ECMO? Is there a difference between the manufacturers?
  • Line 238: Please describe NISHOT with more details.
  • Line 158: Instead of only mentioning LIFEGARDS and EOLIA populations, please provide more details on the numbers. How big were those studies?
  • Line 226: I do not understand this line, what is “ambulating”?
  • Line 268: Anticoagulation: I recommend to give a short statement what current guidelines recommend in terms of anticoagulation and then – based on this information- discuss literature.
  • I would suggest an additional paragraph (“Outlook”) about upcoming innovations and technologies
  • Unfortunately, the VA ECMO is totally ignored in this manuscript, however, in certain situations, VA ECMO is indicated.
  • Line 405: Conclusions: The authors should summarize in more detail the missing data, the main problems to solved with future studies.
  • In times of the COVID pandemic, this manuscript should contain a paragraph about the relevance of V-V ECMO in COVID patients.
  • The reviewer misses take-home messages. The article mentions many topics but does not explain to the reader why these are important.
  • Please provide a table with a list of conclusions
  • Potential complications (implant, bleeding, stroke etc.) should be described more in detail.

Author Response

I would like thank the editors of membranes for the opportunity to review this review article. Giani et al. provide a literature overview of extracorporeal membrane oxygenation in ARDS patients. In times of COVID and the increase of multimorbid patients this topic is of high relevance. A review should allow the reader to get a good overview on the topic. Unfortunately, Giani et al miss to provide all basic information that the average reader needs to understand the article. This review has serious flaws, language has to be improved, it lacks details on the topic and, therefore, needs major revision.  

> We thank the Reviewer for his comment and for his very careful and detailed review. The main Reviewer critique is that our Review does not provide all basic information required by the average reader. We agree with the Reviewer. However, the idea of writing a review for ECMO practitioners was shared with the other two Editors of the “Challenges in the Extracorporeal Membrane Oxygenation Era” special issue (Prof. Arcadipane and Dr. Martucci). Several educational “basic” reviews about ECMO support are available in literature. Therefore, we tried not to linger on basic ECMO information, also considering that our review already counts 4500 words. However, according to the Reviewer comment, more basic information was added in the revised version of the paper.

My comments

In general, the authors mention various relevant topics (ARDS, VILI, NISHOT, ECCO2R etc.) and expect the reader to be up-to-date in these specific fields. I recommend to provide all needed basic information and bring all readers up-to-date before going deep.

> As discussed above, more basic information was provided in the revised manuscript, according to the Reviewer’s comment

  • Which literature did you choose? Please define which studies were included into your literature research (e.g. Top-Journals/prospective randomized trials/retrospective …) and which were excluded.
  • Please add information (+ a figure) of the number of published literature within the last decades since the invention of ECMO

> We included relevant interventional and observational studies and we stated the literature chosen in the introduction. We also added a Table (Table 1) with randomized controlled trials grouped by the review topics

  • I recommend to use another review (about ECMO) from membranes or a comparable journal as “template”

> as discussed above, this was not intended as a basic educational review. This explain the different structure of our manuscript.

  • A short explanation or paragraph of VILI and ARDS should be added to the introduction, since this is the title of the manuscript. In general, the most common indications should be listed as well.

> The explanation of VILI and ARDS has been added to the introduction. The most common indications have been discussed in the “Indications and counterindications for V-V ECMO in ARDS patients” paragraph

  • Although V-V ECMO might be clear to most readers, the introduction would benefit from a short paragraph about the history of ECMO, the difference between V-V- and V-A ECMO and what the most common indications are. Furthermore, a short note about the relevance of V-V ECMO in the current COVID pandemic should be part of the review as well.

> This review does not discuss V-A ECMO. This is now clearly stated in the introduction section.

  • When was the first V-V ECMO described? Why was it developed or who used it if the trials were inconclusive? What were the milestones hat improved the results? Technology?

> The milestones of V-V ECMO in ARDS have been added in the Introduction section.

  • Do not use informal English as e.g. it’s (use it is)

> Language was revised and informal expressions were removed

  • “Moreover, per-protocol and post-hoc Bayesian analysis provided more favor-60 able interpretation of the study results” This information is too rudimental and needs more explanation.

> This was more clearly explained in the revised version of the paper

  • “Indications for V-V ECMO in ARDS patients”: This paragraph reads very unfinished and the text does not flow (general remark). In addition, “Age” cannot be an indication for ECMO, rather a contraindication, therefore it does not correlate to the heading of this paragraph.

> The paragraph was revised. The title was edited as follows “Indications (and counterindications) for V-V ECMO in ARDS patients

  • Line 156: Please define J/min

> Done

  • I would recommend to add a list of abbreviations

> Abbreviations were reduced along the manuscript, however we can provide a list of abbreviations if it is suitable according to the Journal style, this will be discussed with the editorial office

  • All abbreviations must be explained at first use. (e.g. line 183: P/F ratio)

> Done

  • Which implantation techniques are available for V-V ECMO? Is there a difference between the manufacturers?

> The discussion of implantation techniques was beyond the scope of our paper

  • Line 238: Please describe NISHOT with more details.

> For clarity, the expression was replaced with noninfectious transfusion associated adverse events

  • Line 158: Instead of only mentioning LIFEGARDS and EOLIA populations, please provide more details on the numbers. How big were those studies?

> Done

  • Line 226: I do not understand this line, what is “ambulating”?

> The word was edited as follows: “ambulant”

  • Line 268: Anticoagulation: I recommend to give a short statement what current guidelines recommend in terms of anticoagulation and then – based on this information- discuss literature.

> We edited the paragraph according to the Reviewer comment

  • I would suggest an additional paragraph (“Outlook”) about upcoming innovations and technologies

> The paragraph “Outlook” was added to the manuscript

  • Unfortunately, the VA ECMO is totally ignored in this manuscript, however, in certain situations, VA ECMO is indicated.

> The indications to V-A ECMO in ARDS patients have been stated in the Introduction section

  • Line 405: Conclusions: The authors should summarize in more detail the missing data, the main problems to solved with future studies.

> The Conclusions section has been revised and edited according to Reviewer’s suggestions

  • In times of the COVID pandemic, this manuscript should contain a paragraph about the relevance of V-V ECMO in COVID patients.

> The relevance of V-V ECMO in Covid19 patients have been described in the Introduction section

  • The reviewer misses take-home messages. The article mentions many topics but does not explain to the reader why these are important. Please provide a table with a list of conclusions

> The paper has been revised and edited to clarify the take home messages (outlook and conclusions sections)

  • Potential complications (implant, bleeding, stroke etc.) should be described more in detail.

> The discussion of ECMO complications techniques was beyond the scope of our paper, as we decided to focus on management controversies

Reviewer 2 Report

Well written review paper on VV ECMO. Pls go over the paper to ensure appropriate sentence construction.

Author Response

We thank the Reviewer for his/her positive feedback. The paper was revised to ensure appropriate sentence construction.

Reviewer 3 Report

The present review reports on Open Questions, Controversies and Future Directions of veno-venous (VV)-ECMO therapy in acute respiratory distress syndrome.

This is a very well written review, extremely good to read and provides a holistic overview to VV-ECMO.

I got only minor issues to address.

Minor issues:

  1.  
  2. 2.Chapter: I suggest to inform the reader about the reasons for stopping the EOLIA trial early.
  3. From chapter 5 onwards the cumulative numbers of abbreviations are too high for the reader, please could you reduce it
  4. 9.1. Line 289: should be “these” drugs
  5. Line 346: it should be UFH exerts “its”
  6. Please avoid using the abbreviation “RCA”
  7. Could you add a table including randomized trials in VV-ECMO therapy according to the chapter titles of this review

My decision: Minor Revision

Author Response

The present review reports on Open Questions, Controversies and Future Directions of veno-venous (VV)-ECMO therapy in acute respiratory distress syndrome.

This is a very well written review, extremely good to read and provides a holistic overview to VV-ECMO. I got only minor issues to address.

> We thank the Reviewer for his positive feedback. Please find below the answer to his comments.

2.Chapter: I suggest to inform the reader about the reasons for stopping the EOLIA trial early.

> According to the Reviewer's comment, the paragraph was edited as follows:

“The more recent EOLIA trial [19] randomly assigned 249 patients with severe ARDS to receive early V-V ECMO or conventional tidal volume (Vt) and pressure limited ventilation (including late ECMO as rescue therapy). The primary endpoint was mortality at 60 days. However, according to pre-specified futility rules, the trial was stopped early at 75% of recruitment, because of lack of difference in mortality at 60 days between groups. This made the trial probably underpowered to address the research question”.

From chapter 5 onwards the cumulative numbers of abbreviations are too high for the reader, please could you reduce it

9.1. Line 289: should be “these” drugs

Line 346: it should be UFH exerts “its”

Please avoid using the abbreviation “RCA”

              > The paper was revised to reduce the number of abbreviations, according to the Reviewer’s comment. Furthermore lines 289 and 346 were edited according to the Reviewer’s suggestions.

Could you add a table including randomized trials in VV-ECMO therapy according to the chapter titles of this review

              > Done

Round 2

Reviewer 1 Report

I thank the authors for the revision. Although the title suggests the idea of this manuscript, I would recommend to add one line which clearly highlights the scope. 

Minor comments:

Line 62/65: I would not use "will discuss" but "discusses" or "focuses on"

Lin 290-292: I know what the authors want to say, but this sentence is misleading - even for V-V ECMO practitioners. I recommend to rephrase this.

It is problably due to the "track-changes" mode but currently the pdf-manuscript contains the previous and the changed text passages.

Author Response

We thank the Reviewer for his/her positive feedback.

  • I thank the authors for the revision. Although the title suggests the idea of this manuscript, I would recommend to add one line which clearly highlights the scope. 

> The scope of the review was highlighted in the Introduction section

  • Line 62/65: I would not use "will discuss" but "discusses" or "focuses on"

> the verbe tense was modified 

  • Lin 290-292: I know what the authors want to say, but this sentence is misleading - even for V-V ECMO practitioners. I recommend to rephrase this.

> the paragraph was rephrased as follows: 

"A light sedation protocol also allows an early mobilization, which seems feasible and safe in ECMO patients. Indeed, a prospective observational study showed up to 37% of patients with ECMO received some degree of active mobilization. The majority of them was transferred from bed to chair or marched on the spot[85]. In a retrospective study on 100 patients on ECMO support, 35% of patients received active physical therapy with a median delay from ECMO initiation of 2 days. Moreover, 18% of patients ambulated a median distance of 53 meters[86]. Further studies are needed to evaluate if early physical therapy could improve outcomes in ECMO patients, especially during the ECMO-weaning process."

  • It is problably due to the "track-changes" mode but currently the pdf-manuscript contains the previous and the changed text passages.

>the pdf-manuscript was automatically generated by the submission system and contained both the passages.